# Physics-Based TCAD Simulation and Calibration of 600 V GaN/AlGaN/GaN Device Characteristics and Analysis of Interface Traps [note 1]

**DOI:** 10.3390/mi12070751

**Published:** 2021-06-26

**Authors:** Yu-Lin Song, Manoj Kumar Reddy, Luh-Maan Chang, Gene Sheu

**Affiliations:** 1Department of Computer Science and Information Engineering, Asia University, Taichung 41354, Taiwan; manoj14a0kumar@gmail.com (M.K.R.); g_sheu@asia.edu.tw (G.S.); 2Department of Bioinformatics and Medical Engineering, Asia University, Taichung 41354, Taiwan; 3Department of Civil Engineering, National Taiwan University, Taipei 10617, Taiwan; luhchang@ntu.edu.tw

**Keywords:** GaN/AlGaN/GaN, HEMT, TCAD, traps

## Abstract

This study proposes an analysis of the physics-based TCAD (Technology Computer-Aided Design) simulation procedure for GaN/AlGaN/GaN HEMT (High Electron Mobility Transistor) device structures grown on Si (111) substrate which is calibrated against measurement data. The presence of traps and activation energies in the device structure will impact the performance of a device, the source of traps and position of traps in the device remains as a complex exercise until today. The key parameters for the precise tuning of threshold voltage (*V*_th_) in GaN transistors are the control of the positive fixed charges −5 × 10^12^ cm^−2^, donor-like traps −3 × 10^13^ cm^−2^ at the nitride/GaN interfaces, the energy of the donor-like traps 1.42 eV below the conduction band and the acceptor traps activation energy in the AlGaN layer and buffer regions with 0.59 eV below the conduction band. Hence in this paper, the sensitivity of the trap mechanisms in GaN/AlGaN/GaN HEMT transistors, understanding the absolute vertical electric field distribution, electron density and the physical characteristics of the device has been investigated and the results are in good agreement with GaN experimental data.

## 1. Introduction

Gallium nitride (GaN) is one of the superior materials for high frequency and high-power devices for future needs [1,2,3,4,5,6,7,8]. GaN material comes from the III-V group materials which possess the piezoelectric property and spontaneous property in nature, GaN devices such as HEMTs, Metal Insulator Semiconductor HEMTs (MIS-HEMTs) and also Schottky Barrier Diodes (SBDs) are profitable from the presence of large channel charge density (~1 × 10^13^ cm^−2^) at the interface of AlGaN and undoped GaN (Two-Dimensional Electron Gas (2DEG)) region with unintentional doping in the device structure [9,10,11,12,13,14]. GaN HEMT devices have also proven to be the best candidate for operations in critical environments such as high temperature [15,16,17]. This is because of the key features of the device such as wider bandgap, high saturation velocity, very high breakdown voltage and a very good thermal conductivity [14,15,16,17,18,19,20,21]. The optimization of this wideband gap semiconductor devices is still in its early stages and is yet to account for the effect of spontaneous and piezoelectric polarization on the device performance parameters. One such parameter is the Schottky barrier height and its importance for the fact that it relays to breakdown voltage, leakage current and charge control of the device under consideration. GaN device has already confirmed the competence to be the best performing technology for silicon-based power semiconductor devices in power conversions and analog applications.

The interface state density of dielectrics on wide bandgap semiconductors is one of the key performance systems of measurement to measure the quality of dielectrics. In wide bandgap semiconductor based HEMTs, and MISHEMTs the interface between the gate dielectric and the channel is unfavorable to device performance and can cause inefficient Fermi level response and poor gate control. Moreover, when poor-quality dielectrics passivate the access regions, interface states with a relatively long-time constant can lead to current collapse and reduce the maximum current [22,23,24,25].

Electron traps and interface state traps in the structures may have a significant impact on the transfer characteristics of the device. As interface states on the AlGaN layer surface are commonly named and on the other hand traps are in the bulk of the semiconductor. The states localized at the AlGaN and GaN layer interfaces are termed as interface states. It is known from Deep-level transient spectroscopy (DLTS) measurement, that the traps are prevalently located near the AlGaN/GaN interface [26,27].

The advancement of GaN/AlGaN/GaN device has been great throughout the recent decades and now GaN technology is broadly used in industrial-based applications on the silicon substrate. The device physics behind the active defects and traps is not fully understood in GaN/AlGaN/GaN-based HEMTs. So, profound information on the existence and mechanism of the traps and their location is key for the understanding of device transfer characteristics. GaN material has high densities of trap charges which are typically due to crystal imperfections that occur during the growth of the impurities in the lattice, the lattice mismatch between the substrate and crystal lattice, dangling bonds on the surface. Therefore, a precise model or procedure is essential for the calibration of the GaN/AlGaN/GaN device [11]. In the GaN device, the most important cause of electrons in the 2DEG channel is the existence of a fixed charge at the boundary of Si_3_N_4_ and capped GaN. The critical trap positions will vary the electrical performance of the GaN/AlGaN/GaN device and shows the effect on the constancy, stability and consistency of heterostructure devices [28]. 

There are two types of traps present in GaN-based devices namely acceptors and donors. Donor-like trap states can be both positively charged (the possibility to emit an electron) and neutrally charged (when filled). Acceptor-like trap states can be both negatively charged (the ability to capture an electron) and neutrally charged (when empty). In the past, several studies have also investigated the relation between 2DEG and donor states [18]. In these studies, they have shown by what means the 2DEG assets are being controlled by AlGaN barrier layer thickness and its mole fraction when donor-like traps exist in the structure [29,30,31,32,33]. The defects and traps in HEMT devices can be categorized based on the energy level when the trap charges with energy level near to the valence band or conduction bands are known as shallow-level traps. These traps are mainly accountable for parasitic doping effects in the device. Trap charges with the energy levels present deeply in the forbidden bandgap are known as deep-level traps. The procedure of this de-trapping and trapping follows the principles of Shockley Read Hall theory which explains the connections between the free carriers. The key parameters such as fixed charge, the energy level of a donor-like traps, buffer activation energy, barrier height and tunneling coefficient for Schottky gate are key parameters for the understanding of calibration and optimization of HEMT devices.

## 2. Simulation Setup

The GaN HEMT device structure is grown on a silicon (111) substrate with a ~5 µm epi-layer (substrate and buffer) with a 1 × 10^18^ cm^−3^ carbon doping concentration for the buffer layer. The lattice mismatch between the silicon substrate and the AlGaN and GaN buffer stack has been minimized by the introduction of the AlN (Aluminum Nitride) nucleation layer in between the substrate and buffer region. The GaN device structure that is built on silicon substrate serves as the cost-effective device structure [34,35,36,37,38] with good thermal conductivity compared to the GaN substrate itself. In addition to minimizing the lattice mismatch AlN layer can avoid melt-back etching of GaN into Si, can avoid a thermal mismatch between GaN and Si which helps in film preventing cracking and wafer bowing. In buffer stack, Graded AlGaN introduces the optimum compressive stress which compensates tensile stress during cool down to efficiently prevent wafer cracking, and buffer region doped with carbon will provide resistive buffer layer.

GaN undoped layer, AlGaN barrier followed by a tiny deposition of the capped GaN with the Si3N4 passivation layer. The 2DEG region is formed at the interface of the AlGaN barrier and undoped GaN layer. The 2DEG region consists of two charges namely spontaneous polarization for the bond electronegativity and piezoelectric polarization to induce the strain. On top of the AlGaN barrier, a thin GaN cap layer is deposited above the AlGaN barrier layer to reduce the leakage current, current collapse problem and to improve the reliability of the device. The capped GaN HEMT has some advantages over conventional one such as smaller surface roughness, high sheet carrier density and smaller contact resistance. The effective gate width is 100 µm with 5 µm Schottky gate length is also considered for transient simulations [38]. 

The simulated structure of this HEMT consists of a stacked GaN buffer layer, fol-lowed by GaN undoped channel layer, followed by Al_x_Ga_1−x_N barrier layer and GaN cap layer. The device dimensions of the device structure are as follows source to gate distance = 3 µm, Gate length = 5 µm and Gate to drain distance = 20 µm, respectively. Source and drain contacts are ohmic contacts with an annealing temperature of 900 °C for 25 s. Schottky materials are used for the formation of the gate. The entire GaN device structure has been passivated with an oxide (as interlayer dielectric) and nitride (as inter metal dielectric) passivation layer and the isolation device is achieved by nitrogen ion implantation. The conventional device structure by describing each layer is shown in Figure 1.

The charges for GaN/AlGaN/GaN device at each interface are defined as shown in Figure 2 for the physical simulator [39]. The effect of strain relaxation on the piezoelectric charges was not accounted for simulation [40]. This strain relaxation would not affect the main moto of the paper which is to study the impact of donor-like traps and fixed charges on the HEMT characteristics. The fixed charges which were given at the interface of Si_3_N_4_ and GaN interface (*σ*_2_) were assessed by considering the trap charges related to the passivation layer composed with the piezoelectric polarization charges. Donor-like traps (*σ_D_*) were also given at the same interface (Si_3_N_4_/GaN) [29,30,36] with some donor-like traps energy level.

## 3. Results and Discussion

A HEMT is a heterostructure device with two different layers in which narrow bandgap material is grown first followed by the wide bandgap material. The band diagram of conventional GaN/AlGaN/GaN HEMT device structure is picturized in Figure 3. 

From Figure 3 the location of the quantum well is observed at the boundary of the AlGaN barrier region and GaN undoped region as a manifestation of piezoelectric and spontaneous effect. From this simulation, the confined 2DEG concentration can be observed with a concentration around 2 × 10^19^ cm^−3^. The 2DEG concentration is in good agreement as reported [11].

Before the calibration of parameters, one has to select the best model that is available in the TCAD simulation software. For the current GaN HEMT device, the models used are as shown in Table 1 [41,42]. The high field saturation model will explain the presence of high critical electric fields. The avalanche generation model explains that Electron–hole pair formation is due to the avalanche generation in the device structure or the impact ionization process needs a firm threshold field point a. The recombination process over the deep defect points in the gap is classically known as Shockley Read Hall’s theory for the recombination process.

In this study, the investigation on the position of donor-like traps which are present at the passivation (Si_3_N_4_ layer) and top-layer interface (GaN cap layer) using a single trap energy level than continuously distributed states is carried out. With the understanding of the location of the trap and their energy levels, we have explained the device characteristics such as breakdown voltage, the threshold voltage (*V*_th_) and *I*_d_
*− V*_d_ results. This work has been carried out by using the TCAD simulation tools.

In this study, few electrical parameters including *I*_d_
*− V*_g_, *I*_d_
*− V*_d_, off-state bias and dynamic on-resistance (R_ON_) are measured and calibrated against GaN wafer data. The measurement conditions are described in Table 2. In all measurement conditions, the substrate terminal will be set to the floating condition. It is found that the floating substrate termination not only enables higher OFF-state breakdown voltage but also delivers the benefit of smaller dynamic R_ON_ degradation and output capacitance under the drain bias of over 400 V for the switching operations [43,44,45]. There are four key parameters required to calibrate against GaN/AlGaN/GaN HEMT device. They are discussed in this section as follows.

### 3.1. Schottky Barrier Height

In this paper, the simulation results carried out with Schottky barrier height value of 0.7 eV are in good agreement with the GaN wafer experimental data and the sensitivity of the barrier height on threshold voltage (*V*_th_) and gate leakage current is shown in Figure 4. During the epitaxial layer, growth defects will be formed in the device [46]. The defect charges present in the device structure will also affect the breakdown voltage and advances the impact ionization coefficient due to the presence of an electric peak field at the edge of the gate electrode on the drain side [35].

### 3.2. Tunneling Coefficient

The electrons that tunnel from the gate electrode may generate a leakage current at the gate to drain by bouncing from one trap charge to the other. 

The tunneling component mechanism is very significant for GaN device besides the Poole-Frenkel (PF) discharge current which governs in comparatively lesser negative bias [47]. The higher the tunneling coefficient the more leakage current as shown in Figure 5. According to Figure 5, the change in the tunneling coefficient will not affect the threshold voltage (*V*_th_) or drain current level since the gate leakage current is allocated to Poole-Frankel emission and non-local tunneling under the gate region. For this device, the 0.185 tunneling coefficient gives the best result for gate leakage current calibration.

### 3.3. Fixed Charge and Donor-Like Traps

The charges which are assumed at the boundary of Si_3_N_4_ and GaN are fixed charges (*−σ2*) as shown in Figure 2. The positive trap charges will gradually increase the peak electric field at the AlGaN barrier layer under the gate region and the higher coefficient of tunneling allows electrons to tunnel deeper into the AlGaN barrier layer while a higher |V_G_| shifts the conduction band higher which weakens the tunneling barrier and increases reverse gate characteristics [42]. Donor traps are uncharged when unoccupied and they carry the charge of one hole when fully occupied. The fully occupied donor traps will cancel out the effect of negatively bias gate voltage. In our study, it is found that a fixed charge of −5 × 10^12^ cm^−2^ and donor traps of 3 × 10^13^ cm^−2^ gives the best fit of the simulation data with the GaN experimental data. [10,13,29,35,36,41,47,48].

### 3.4. Donor and Acceptor Energy Levels

The fitting procedure for donor energies yields a value of 1.42 eV which agrees with the GaN wafer experimental data. Thus, the assumption of the existence of donor-like traps located at 1.42 eV below the conduction-band of the Al_x_Ga_1-x_N barrier layer has been justified. The density of surface donor traps is at least 1 × 10^13^ cm^−2^ for the 2DEG density. In addition to the donor energy, the activation energy for the carbon-doped buffer region for the acceptor traps should also be 0.59 eV below the conduction band [29,35,36,49] to attain a positive temperature coefficient [50].

Figure 6 shows the breakdown voltage calibration of the GaN/AlGaN/GaN device by using the van overstraeten model and Figure 7 shows the vertical electric field distribution for the same device. Since the GaN/AlGaN/GaN HEMT device can be grown on various substrates, the impact ionization coefficient calibration is very important to reproduce an avalanche breakdown point of the device as shown in Figure 8 [51]. In this experiment, the device is built for the breakdown voltage of 600 V and the impact ionization model is ruled by using the van overstraeten model. van overstraeten’s model, which is based on Chynoweth’s law [52]. In this study, the breakdown voltage principle working mechanism for GaN/AlGaN/GaN HEMT shows that the avalanche point is due to the impact ionization [53].

The peak electric field present in the GaN HEMT device structure can be observed from two spots mainly they are gate edge at drain portion and gate field plate edge as shown in Figure 7. The cutline is drawn along the X-axis in the 2DEG channel region at the interface of the AlGaN barrier layer and GaN undoped layer. The vertical component of the electric field in the AlGaN barrier layer is representative for the electric field strength in the barrier (where the vertical component is constant, and vertical peak field is much larger than the horizontal one). This vertical electric peak field can be optimized by varying the length of the gate field plate. By minimizing the vertical electric peak field distribution at the gate edge on the drain side helps suppress the current collapse and improves the dynamic Ron ratio. This peak reduction field will also help in improving the electrical characteristics as discussed below.

Figure 8 clearly shows that breakdown voltage can be enhanced by introducing the gate field plate. The reduction of the electric peak field at the gate edge has helped to improve the breakdown voltage of the device. From the curves, without field plate, GaN HEMT device has 1115 V breakdown voltage which is less than with field plate GaN HEMT device. With the introduction of the gate field plate, the breakdown voltage is 1574 V which showed 40% improvement and the obtained curve is avalanche breakdown mechanism which is improved due to the impact ionization.

Figure 9 shows the calibration of the *I*_d_
*− V*_g_ characteristics curve simulated result comparison with GaN wafer experimental data at room temperature. The drain current calibrated results show remarkably very good agreement with the GaN wafer experimental data. The results are in good agreement due to the calibration of key parameters of barrier height, tunneling coefficient and trap charges. 

The drain current stays comparatively saturated with an additional negative sweep of the gate electrode (the drain leakage current is measured in OFF state condition). As soon as positive ionized donors compensate the fixed trap charges, an increment in the donor trap concentration will subsidize the electrons to the channel region. Meanwhile, the GaN wafer experimental data has also shown that the gate leakage current will rise with a comparable amount of the total drain current. This indicates that gate current is being dominated by the leakage current between drain-gate electrodes. The threshold voltage (*V*_th_) point for the data is considered at the drain current level of 1 mA/mm with a drain voltage of 20 V. For the presented device structure, the device width is 100 µm, the drain current level for determining the device on/off state is 1 × 10^−4^ A and the measured threshold voltage (*V*_th_) at that point is −3.4 V.

The calibrated *I*_d_
*− V*_d_ with various gate voltage at room temperature is as shown in Figure 10. The simulated results are in good agreement with the GaN wafer experimental data. Pulsed *I*_d_
*− V*_d_ measurements were carried out at several bias points to compute primary classification and to realize which trapping mechanisms may affect the transistor’s behavior. In the simulated results, the current degradation is not pronounced at lower gate voltages (at *V*_g_ = −3 V and *V*_g_ = −4 V) whereas the GaN wafer experimental data is showing the current degradation at *V*_g_ = −3 V and *V*_g_ = −4 V. The effect of current degradation is probably triggered by numerous factors together with the self-heating effect and current collapse phenomenon. Nevertheless, it is stated in [54,55] that the primary reason for the current collapse is the presence of bulk traps in the carbon-doped buffer region and the energy levels of the acceptor traps.

The gate electrode edge on the drain electrode side of the results has shown a high electric peak field and comparatively higher temperature, which will reduce the carrier mobility property of the device and increases the current collapse behavior. The impact ionization coefficient will determine the key characteristics of the GaN HEMT device [56,57]. The impact ionization concentration distributions confirms the inference and a point to note is that the electron-hole pair production caused by the avalanche breakdown entails the threshold field strength and the opportunity of acceleration, that is, a wide space charge region. 

The key point here is that the impact ionization peak typically doesn’t occur at the peak point of the electric field because the field positions of devices are different and the dispersals of space charge regions are also different. This is the reason for possible inconsistency between the electric field and impact ionization concentration distributions [58]. The advantage of using the TCAD simulations is we can easily identify the high peak electric field which is located at the gate edge and gate field plate edge on the drain side as shown in Figure 11. Based on the literature study [34], this peak electric field located at the edge of the gate region and gate field plate region will increase the possibility of trapping occurrences which will affect the deterioration of the dynamic Ron ratio.

Figure 12 explains the mechanisms of source-drain current modulation. More negative gate voltage will repel the 2DEG under the gate and the other possible mechanism is the weakened piezoelectric effect which induces less 2DEG concentration [58,59,60]. The cutline is drawn horizontally along the X axis in the AlGaN barrier layer. From the simulation, the critical region during negative gate bias can be found in the gate edge at the drain side. This spot shows a relatively high vertical peak electric field. The high electric field at the gate edge has been reported to be one of the factors that causing current collapse. The source to drain current is modulated by the applied gate voltage.

The important pattern that can be observed based on the picture above is 2DEG concentration will be inversely proportional to the vertical electric field value. This pattern tells us that modification on this peak electric field will change the value of the threshold voltage (*V*_th_). In this case, the vertical electric field value has been normalized into absolute value. Originally it has a negative value which indicates that vertical electric field direction is to the surface.

Figure 13 is showing the sensitivity test of *I*_d_
*− V*_d_ at *V*_g_ = 0 V with various boundary thermal resistance at the substrate. The initial temperature of the substrate boundary is 300 K. The trend shows that the self-heating effect reduces the drain current when the drain voltage is biased more than 5 V with gate voltage equals to 0 V. This calibration is very important for Ron calibration. Self-heating simulation is critical to show the temperature hot spot in the device [61,62,63]. 

From Figure 14, the gate edge of the drain side is showing a relatively high temperature. The high temperature will degrade carrier mobility, and this phenomenon is well observed by current degradation as depicted by Figure 13 for the GaN HEMT device. The thermal conductivity model applied to the simulation can be expressed as follows:k(*T*_L_) = (TC.CONST)/(*T*_L_/300)^TC.NPOW^
where TC.CONST is a thermal conductivity constant of each material for 300 K and TC.NPOW is an experimental value of each material for the thermal conductivity model. 

## 4. Conclusions

From this study, simulation results have been calibrated against the GaN wafer experimental data which are in good agreement as presented in the form of breakdown voltage, *I*_d_
*− V*_g_ and *I*_d_
*− V*_d_ curves. In this study, we have found that the trap charges will play a vital role in understanding the GaN HEMT device analysis, modeling and understanding its characteristics. However, some of wafer experiment setups, traps and defects properties from several literature studies and the calibration results against the GaN wafer experiment data has identified that the amount of fixed charge traps is −5 × 10^12^ cm^−2^ present at the interface of (Si_3_N_4_/GaN) and the total donor trap charges is 3 × 10^13^ cm^−2^ with the energy level of 1.42 eV below conduction band energy. The self-heating effect has been adopted to predict the saturation current. Even though the mechanism of the GaN device breakdown is not fully understood, the impact ionization coefficient has been considered to accommodate the GaN material breakdown properties grown on the silicon substrate. By analyzing the transfer characteristics of the GaN HEMT device structure we have characterized the trap charges that resemble those located at the surface in the drift region of GaN/AlGaN/GaN HEMT.

Eventually, we have understood that with the proper usage of TCAD simulation methodologies the TCAD tools will help improve the characteristics and reliability of the device. The critical areas present in the device structure can be identified with the help of simulated results, one of such critical areas is located at the gate electrode edge of the drain electrode side which has a hot spot of high peak electric field, high impact ionization and high lattice temperature.

## Figures and Tables

**Figure 1 micromachines-12-00751-f001:**
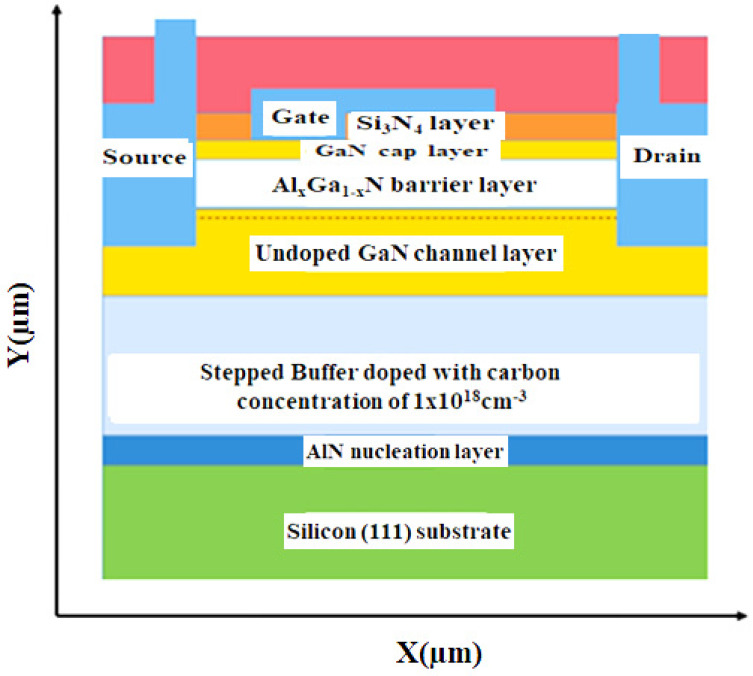
The conventional device structure of GaN/AlGaN/GaN HEMT.

**Figure 2 micromachines-12-00751-f002:**
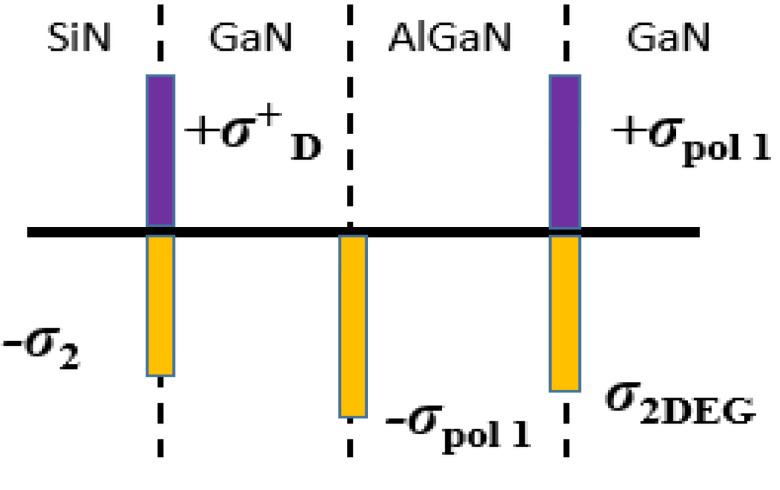
Polarization charges defined for GaN/AlGaN/GaN device in TCAD simulation.

**Figure 3 micromachines-12-00751-f003:**
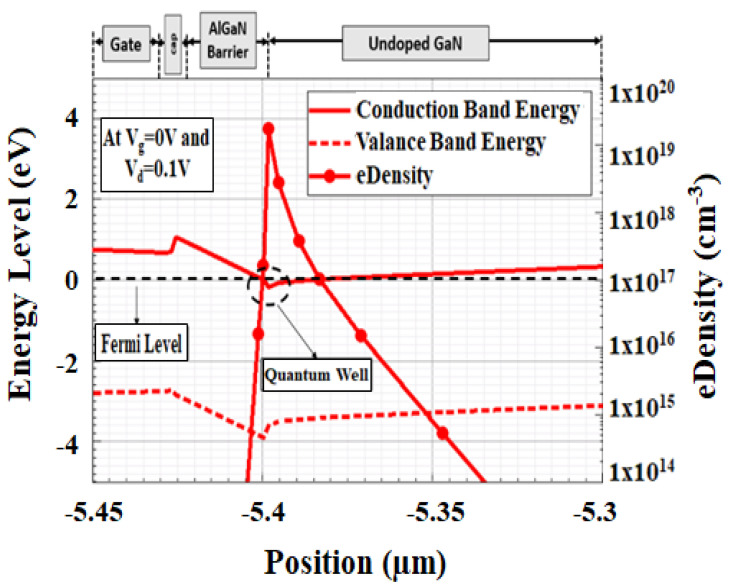
Simulated energy band diagram and electron density for GaN/AlGaN/GaN HEMT device by drawing the vertical cutline along the device.

**Figure 4 micromachines-12-00751-f004:**
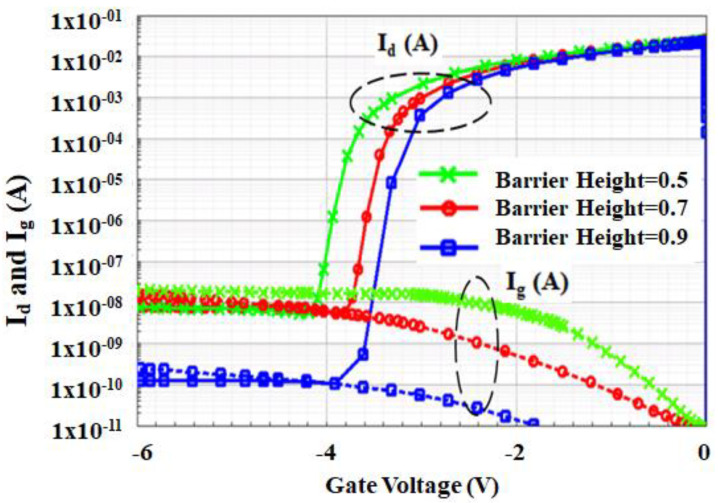
Simulated impact of Schottky barrier height on Id and Ig.

**Figure 5 micromachines-12-00751-f005:**
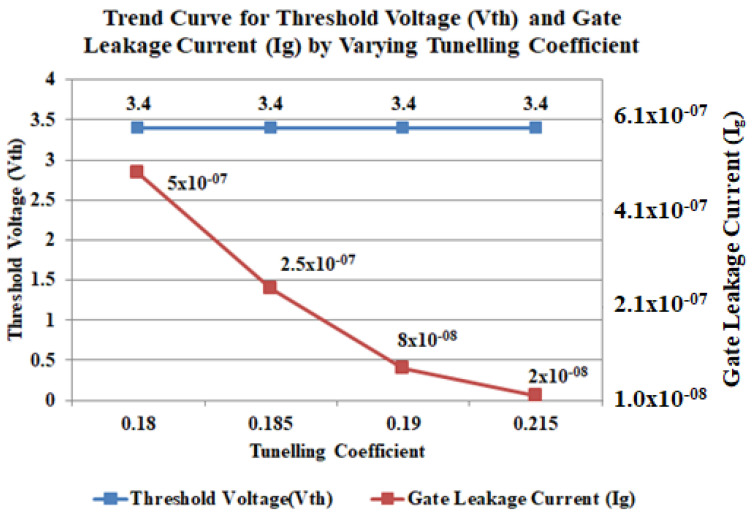
Simulated points on the impact of tunneling coefficient on threshold voltage (*V*_th_) and gate leakage current.

**Figure 6 micromachines-12-00751-f006:**
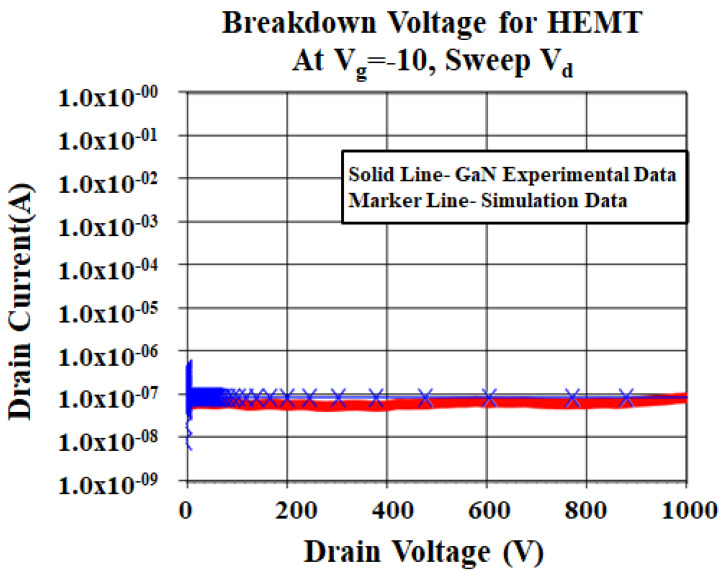
Breakdown voltage curve of the simulated data and GaN wafer experimental data.

**Figure 7 micromachines-12-00751-f007:**
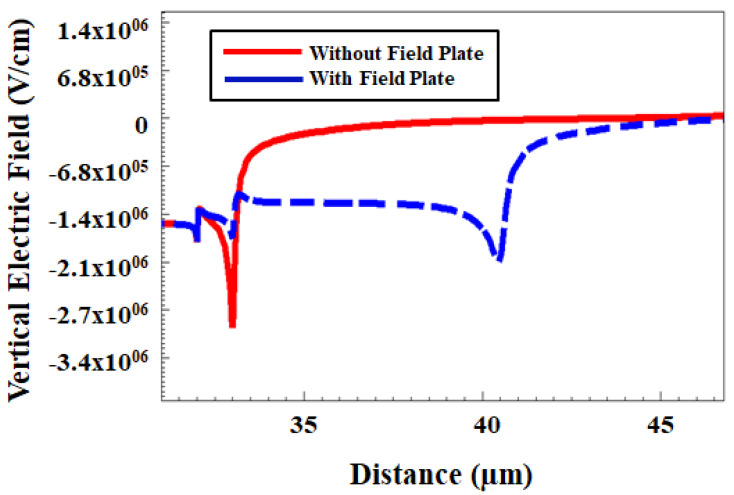
Simulated vertical electric field distribution with gate field plate and without gate field plate at 600 V breakdown voltage by drawing the cutline along X-axis in the 2DEG channel region.

**Figure 8 micromachines-12-00751-f008:**
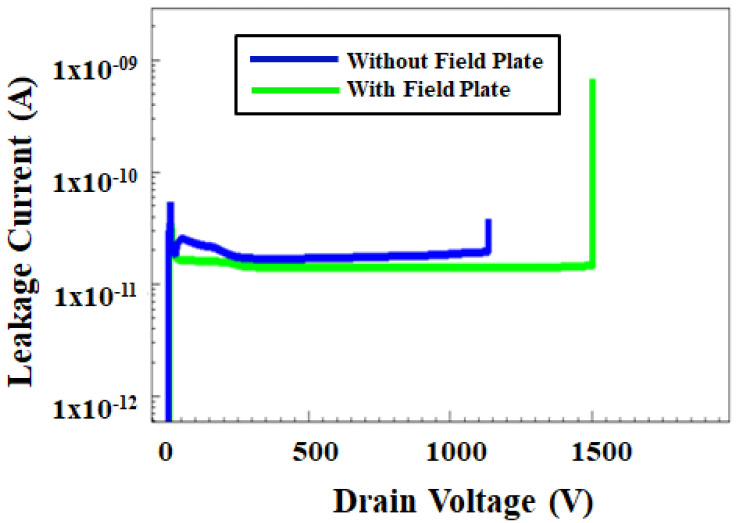
Simulated breakdown voltage comparison with gate field plate and without gate field plate.

**Figure 9 micromachines-12-00751-f009:**
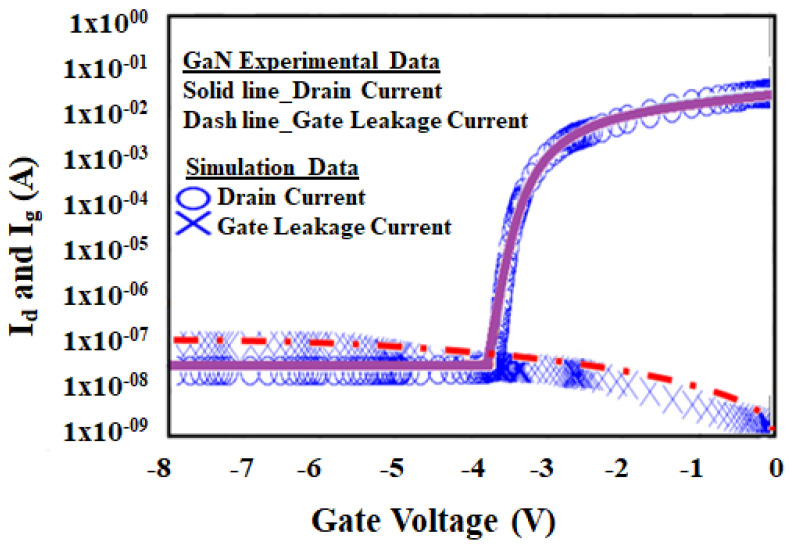
*I*_d_*− V*_g_ characteristics comparison of simulated data and GaN wafer experimental data.

**Figure 10 micromachines-12-00751-f010:**
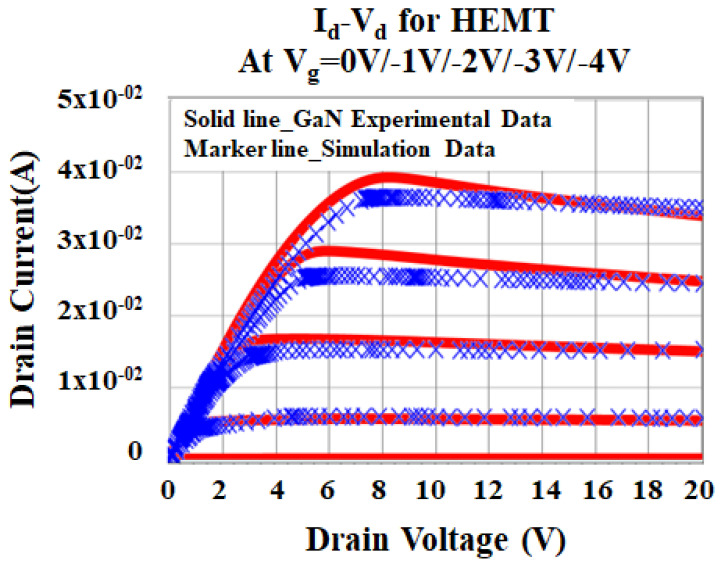
*I*_d_*− V*_d_ characteristics of the simulated data and GaN wafer experimental data.

**Figure 11 micromachines-12-00751-f011:**
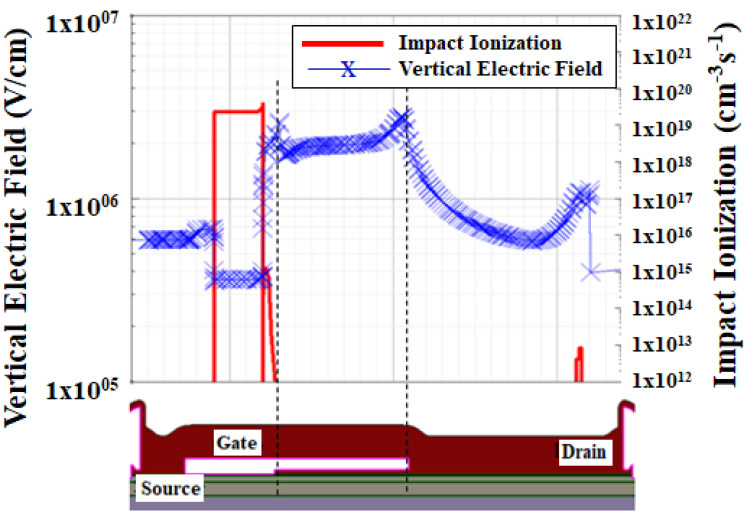
Simulated Impact ionization and vertical electric field distribution along the X-axis at 2DEG region in GaN HEMT device.

**Figure 12 micromachines-12-00751-f012:**
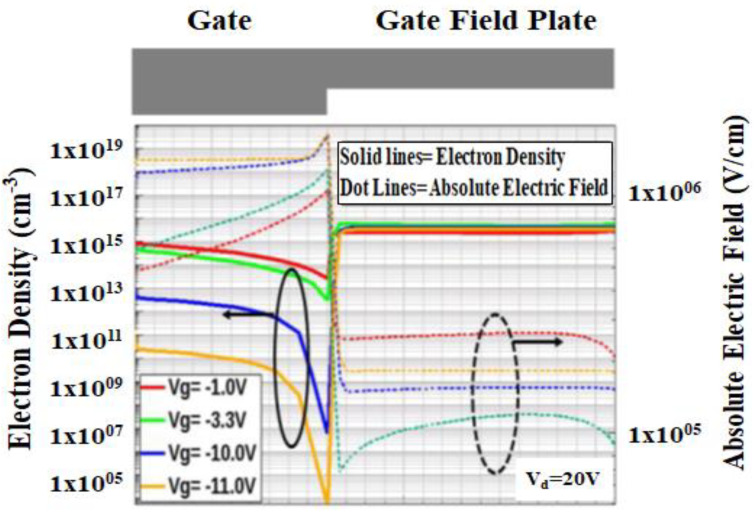
Simulation of electron density and absolute vertical electric field distribution in GaN HEMT device.

**Figure 13 micromachines-12-00751-f013:**
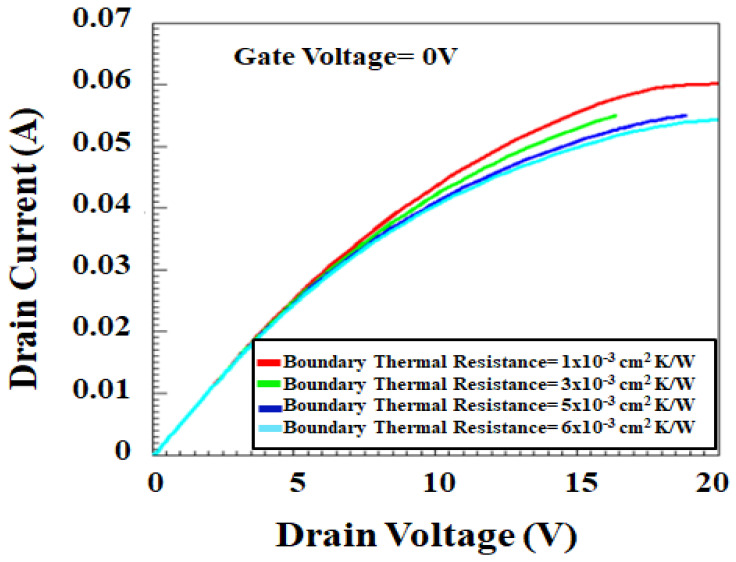
Simulated *I*_d_
*− V*_d_ based on self-heating mode for various thermal resistance.

**Figure 14 micromachines-12-00751-f014:**
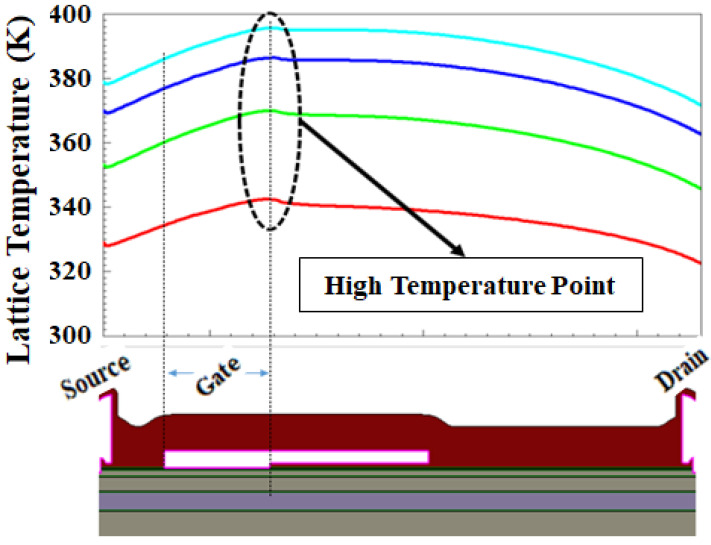
Simulated lattice temperature distribution at the interface of AlGaN and GaN undoped region with horizontal cutline.

**Table 1 micromachines-12-00751-t001:** Models used in GaN device simulation.

Physical Phenomenon	Model
1. Mobility	1a. Doping dependence
	1b. High field saturation
	1c. Poole frankel
2. Avalanche	2a. Van overstraeten
3. Recombination	3a. Shockley-Red-Hall
4. Polarization	4a. Piezo-Electric Stress
	4b. Piezo-Electric Strain
5. Tunneling	5a. Electron Barrier Tunneling
6. Self-heating effect	6a. Thermodynamic

**Table 2 micromachines-12-00751-t002:** Measurement parameters and conditions.

Parameters	Conditions
Threshold voltage (*V*_th_)	*V*_GS_ at *I*_D_ = 1 mA/mm
*I* _Dss_	At *V*_GS_ = 0 V; and at *V*_DS_ = 20 V
Off-state Breakdown	At *V*_GS_ = *V*_th_ *−* 5 V; *V*_DS_ sweep
Dynamic R_ON_ [30]	At *V*_GS_ = *V*_th_ *−* 5 V; *V*_DS_ stress = varied Off-state period = 80 µs; On-state period = 10 µs

## Data Availability

Not applicable.

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
