# Peer review of "Physics-Based TCAD Simulation and Calibration of 600 V GaN/AlGaN/GaN Device Characteristics and Analysis of Interface Traps [Author-notes fn1-micromachines-12-00751]"

_micromachines, 2021, doi:10.3390/mi12070751_

Round 1

Reviewer 1 Report

In this paper, the authors simulated the AlGaN/GaN HEMT on silicon substrate by TCAD. The performance simulated includes the breakdown voltage, IdVd , IdVg and threshold etc.  The traps effect on the electrical properties was also discussed and simulated.  The comparison with experiments was also conducted. This paper is timely important for the development of GaN-based power and rf devices. Here are the comments that need to be addressed by the authors.

(1) In Fig.5, what is the red part?

(2) Ln 94, 5um is 4um?

(3) What is the Schottky metal?

(4) Is SiN or Si3N4?

(5)  It will be better to show the traps location/densities in an additional figure.

(6) Similar simulation  was also reported in the literature of  "2D study of AlGaN/AlN/GaN/AlGaN HEMTs’ response to traps" (doi: 10.1088/1674-4926/40/2/022802). The authors should  refer this work for comparison. 

Reviewer 2 Report

The reviewed manuscript presents interesting studies on the TCAD simulation of GaN/AlGaN/GaN high electron mobility transistor. The paper can be accepted in the Micromachines journal after the revision of the following issue.

The authors in the TCAD simulations applied a physical model of the interface charges (Fig. 2), which included, in particular, the donor-like fixed charge and interface state charge at the SiN/GaN interface (-sigma2). The charge related to the  interface states at the SIN/GaN interface is fixed and equal to the donor like-fixed charge. However, in the general case, the interface states at the insulator/nitrides exhibited a broad  continuum distribution in the band gap and thus the interface- state occupation function should be taken into account. I do not expect from the authors to perform a new simulation with the interface-state occupation function but as the minimum this issue in the paper should be commented.  The charge distribution in the  SiN/AlGaN/GaN heterostructures which included the continuum interface  trap distribution you can find in  M. Matys <https://aip.scitation.org/author/Matys%2C+M>  et al  Appl. Phys. Lett. 110, 243505 (2017). Furthermore, more about the connection between occupation function and the threshold voltage in insulator/GaN/AlGaN/GaN heteresturctres can be found in M. Ťapajn <https://aip.scitation.org/author/%C5%A4apajna%2C+M> a et al. Journal of Applied Physics 116, 104501 (2014). Please comment the interface state occupaction and include these two papers in References.

Reviewer 3 Report

The manuscript presents a simulation study of AlGaN/GaN HEMTs performed with TCAD. Even though the work is technically sound, its contribution is of very low value and this reviewer recommends to reject. The main reasons are the following:

  1. The work focuses on showing TCAD simulations of very well known aspects of AlGaN/GaN HEMTs design, which do not contribute to the knowledge of the community, i.e. barrier  height, field plates and self-heating have been studied in depth before and no significant contribution is presented here.
  2. The treatment of traps is very shallow and does not offer any physical insight on the mechanisms of trap formation or passivation, and many other studies already published offer much more useful information. 
  3. Unfortunately and with all due respect, this reviewer feels that at some points the language is confusing or incorrectly used and significant improvement is required. 

For the above reasons, I recommend to reject the manuscript.

Round 2

Reviewer 1 Report

The paper can be accepted now.

Reviewer 3 Report

The reviewer appreciates the author's efforts to improve the manuscript. Nevertheless, after careful consideration I recommend to reject it for publication. The main reason is the negligible contribution of the results to the state of the art knowledge in AlGaN/GaN HEMTs technology. 

in particular:

  1. The authors use TCAD to simulate an AlGan/GaN HEMT and claim they offer insight on the effect of traps. However, all they do is add a fixed charge layer, which has been done for decades in the simulation world),to fit a single device. The particular values reported are simply a fitting parameter for this particular case.  This is strategy is very well known, very old and it does not contribute to the community. To claim the manuscript offers insight on the effect of traps is just an unsubstantiated overstatement. 
  2. The authors provide the systematic variation of some parameters in their simulations to show the impact on the IDVG curves. This is is not an original study and does not contribute to the community. The effect of schottky barrier height, field plates, VT and leakage, on IV curves are all well understood. Any of the authors references offer a deeper insight in any of these topics than the manuscript under review. 
  3. The reviewer understands that learning how to use a software tool like  TCAD is not easy and it requires some effort and understanding. However, the author's self improvement and discovery of well known effects are simply not enough for publication in this journal.

Author Response

This manuscript is a resubmission of an earlier submission. The following is a list of the peer review reports and author responses from that submission.

Round 1

Reviewer 1 Report

see attached file.

Reviewer 2 Report

There are a few things in the manuscripts that may be revised:

  1. In line 93, the authors should revise that the strain is the reason for piezoelectric polarization, but not vice versa.
  2. Throughout the manuscript, the meaning of the negative density of fixed charges, such as -5x1012 cm-2, is not understood.
  3. Please specify the substrate in line 49 instead "GaN/material".
  4. Please provide the thicknesses of each layer on the schematic sample structure in Figure 1. Also indicate the material for the buffer layer.
  5. An extensive language editing is required.

Round 2

Reviewer 2 Report

Thank you for considering and answering fo the previous comments.